# A Sensor-Driven Analysis of Distributed Direction Finding Systems Based on UAV Swarms

**DOI:** 10.3390/s19122659

**Published:** 2019-06-12

**Authors:** Zhong Chen, Shihyuan Yeh, Jean-Francois Chamberland, Gregory H. Huff

**Affiliations:** 1Department of Electrical and Computer Engineering, Texas A&M University, College Station, TX 77843-3128, USA; zhongchen@tamu.edu (Z.C.); steven.yeh66@gmail.com (S.Y.); chmbrlnd@tamu.edu (J.-F.C.); 2School of Electrical Engineering and Computer Science, Pennsylvania State University, University Park, PA 16802, USA

**Keywords:** direction-of-arrival estimation, unmanned aerial vehicles, UAV swarm, aperiodic arrays, MUSIC, Cramer–Rao bound

## Abstract

This paper reports on the research of factors that impact the accuracy and efficiency of an unmanned aerial vehicle (UAV) based radio frequency (RF) and microwave data collection system. The swarming UAVs (agents) can be utilized to create micro-UAV swarm-based (MUSB) aperiodic antenna arrays that reduce angle ambiguity and improve convergence in sub-space direction-of-arrival (DOA) techniques. A mathematical data model is addressed in this paper to demonstrate fundamental properties of MUSB antenna arrays and study the performance of the data collection system framework. The Cramer–Rao bound (CRB) associated with two-dimensional (2D) DOAs of sources in the presence of sensor gain and phase coefficient is derived. The single-source case is studied in detail. The vector-space of emitters is exploited and the iterative-MUSIC (multiple signal classification) algorithm is created to estimate 2D DOAs of emitters. Numerical examples and practical measurements are provided to demonstrate the feasibility of the proposed MUSB data collection system framework using iterative-MUSIC algorithm and benchmark theoretical expectations.

## 1. Introduction

DOA estimation of source using array sensors plays an important role in the field of array signal processing. DOA estimation includes one-dimensional (1D) DOA estimation (azimuth) and 2D DOA estimation (azimuth and elevation). The accuracy of DOA estimation is mainly impacted by the algorithm, the geometry of sensor array structure, signal-to-noise (SNR), and snapshots, etc. Algorithms and array geometry are two essential research topics.

The DOA estimation method has been studied for decades and it is still an active research topic in recent years. Initially, DOA estimation based on sensor array structures used the Bartlett beamforming method, but it cannot offer high-resolution due to the Rayleigh limit [1,2]. Then, Burg came up with the maximum entropy (ME) method, which is a high-resolution method, but it has a low robustness and a considerable computation [3]. Later, a series of high-resolution spatial spectrum estimation methods based on decomposition of matrix eigenvectors for array signal processing came out and created a new era. All those spatial spectrum estimation methods were represented by MUSIC and estimation signal parameters via rotational invariance technique (ESPRIT) [4,5,6,7]. These two methods have greater resolution and accuracy than other classical methods. The simulations in [8] indicate that MUSIC algorithm is more accurate and stable than ESPRIT algorithm for uniform linear array (ULA). Furthermore, ESPRIT can only be used for invariant geometry, while MUSIC can be applied for arbitrary non-uniform sensor arrays and multiple-source estimation. The MUSIC algorithm attracted more and more attention since it came out, which was the milestone of spatial spectrum estimation methods. In certain conditions, MUSIC algorithm is 1D implementation of maximum likelihood (ML) method, which shares the same characteristic with ML [9,10]. Then, due to the relatively high calculation complexity of MUSIC, some search-free algorithms, such as root-MUSIC, manifold separation based on root-MUSIC, use the root-solving technique to reduce the computational complexity [11,12]. Nevertheless, it can only be used for ULA. In order to deal with arbitrary non-uniform arrays, the Fourier domain root-MUSIC (FD root-MUSIC) algorithm was addressed [13], but the FD root-MUSIC algorithm is mainly used for 1D DOA estimation.

Most of algorithms above are still in simulation and theoretical stages. Recently, practical implementation of the DOA estimation system using field programmable gate array (FPGA) and digital signal processing (DSP) are reported [14,15,16], those implementations achieve real-time application of DOA estimation based on the MUSIC algorithm. Different from other well-known subspace-based techniques like ESPRIT, MUSIC algorithm has many advantages in the real implementation owing to its simplicity and suitability for parallel processing [16]. 

Apart from DOA estimation methods, the geometry of the antenna array is also very important. Most of the investigations on the array structure are limited in either 1D linear array or 2D planar array configurations [17]. The 1D array can only detect 1D DOA, and 2D array structures such as uniform rectangular array (URA) and uniform circular array (UCA) can well estimate azimuth angles but cannot well estimate elevation angles due to its small antenna aperture in the elevation direction. In order to improve the elevation angle estimation accuracy, one may develop a three-dimension (3D) array structure by putting more elements in the vertical direction to make a large elevation aperture [18,19]. However, it requires additional hardware and computational cost. Furthermore, the uniform linear and planar array will cause the ambiguity problem (angle aliasing) due to the symmetry of array structures [20,21,22]. Xia et al. proposed that cubic arrays still have ambiguity problem and the spherical array can significantly reduce this problem [22]. Recently, 3D antenna array configurations have attracted much more research interest in array signal processing [22,23,24,25,26,27,28]. Most of those 3D arrays above are constructed from regular structure (i.e., cubic, cylinder), extending the planar array (i.e., URA, UCA) or configuring the virtual 3D array based on the planar array. Even though those special 3D arrays increase the elevation angle estimation accuracy, their array apertures are still relatively small since the physical size of static arrays are restricted. Moreover, conventional investigations in the DOA estimation require that the number of sensors is more than the number of receiving signals, which increase hardware cost and system complexity.

In contrast, in order to investigate the compromise between hardware cost and signal processing time, time-variant arrays whose element positions are changed over time are examined by time-divided sampling rather than simultaneously sampling as static array. Many researchers have reported using time-variant arrays to improve DOA estimation performance [29,30,31,32,33,34,35]. Instead of using a set of different elements to process the incident signals, the time-variant array can only use one or a small number of moving elements to construct virtual antenna arrays. Wan et al. proposed a method of combining the characteristics of arbitrary virtual baseline to construct virtual 3D array [29]. However, the number of sub-array elements is too small so that it cannot have high resolution. Liu examined a rotating long baseline interferometer whose length is much larger than one wavelength to estimate 2D DOAs by constructing virtual 2D circular arrays [30]. However, the 2D circular array has limited elevation aperture and still cannot well estimate the elevation angle. 

The MUSB aperiodic array reconstructed from swarming UAVs proposed herein has aperture dimensions in both azimuth and elevation directions, which increase the accuracy of both azimuth and elevation angle estimation. Corner and Lamont proposed a parallel simulation of UAV swarm scenarios [36], and Saad et al. reported a testbed of vehicle swarm rapid prototyping [37]. Recently, many researchers investigated the methods and impact factors of designing the robust MUSB antenna arrays for signal collection platforms [38,39,40,41]. However, those works are limited in MUSB array constructing investigations including UAV positional precision, turbulence of the environment, micro-UAV swarm algorithm, and swarm-based real-time data collection. In this paper, 2D DOA estimation using the MUSB aperiodic array is provided, a mathematical model of the MUSB data collection system for signal processing is offered firstly and the impact of associated parameters on DOA estimation accuracy and convergence in this model are analyzed. The MUSB arrays have advantages of large aperture, large interelement spacing, no shadowing effect, low mutual coupling effect and large spatial sampling data from different locations in free space. 

In practice, the MUSB system for DOA estimation requires low snapshots and might be applied in low SNR scenarios. However, the subspace-based techniques require adequate SNR and snapshots to guarantee good performance. We utilize the iterative method to lower the noise floor by multiplying the current MUSIC spectrum and previous spectrum for each iteration. The details of the iterative-MUSIC algorithm will be presented in Section 5. 

This paper mainly contributes to the array signal processing area in three aspects. First, the mathematical model of the MUSB aperiodic array data collection system is introduced to demonstrate the fundamental DOA estimation impact factors and performance. Second, the CRB associated with DOAs in the presence of the gain and phase coefficient in the system is derived to reveal some direction-finding properties such as the global convergence, snapshots, SNR, and the number of arrays. Third, a successive DOA refinement procedure with iterative-MUSIC algorithm is provided based on the reconstructed arrays and spectrum to meet the requirement of high-precision DOA estimation. The rest of this paper mainly consists of six sections. Section 2 introduces the MUSB mathematical model; Section 3 derives the CRB associated with source DOAs in the presence of the sensor gain and phase coefficient; Section 4 proposes performance analysis of the MUSB system for the single-emitter case; Section 5 gives the algorithm applied in this paper; Section 6 provides simulation and measurement results in different scenarios and Section 7 concludes the paper.

Glossary of notations is listed below: $C^{k\times p}$ = the space of k×p complex-valued matrices; *E* = expectation operator; $A_{ij}$ = the i, j element of a general matrix $A\in C^{k\times p}$; $A^{T}$ = the transpose of $A\in C^{k\times p}$; $A^{H}$ = the conjugate transpose of $A\in C^{k\times p}$; Re (A) = the real part of $A\in C^{k\times p}$; Im (A) = the image part of $A\in C^{k\times p}$; tr (A) = the trace of $A\in C^{k\times k}$; det (A) = the determinant of $A\in C^{k\times k}$; A⊙B = the Schur–Hadamard matrix product of $A,B\in C^{k\times p}$, defined by ${\left[A⊙B\right]}_{ij}=A_{ij}B_{ij}$; A⊗B = the Kronecker matrix product of $A,\;B\in C^{k\times p}$, defined by
(1)$$A⊗B=\left[\begin{array}{ccc} a_{11}B & \cdots & a_{1j}B\\ \vdots & \ddots & \vdots \\ a_{i1}B & \cdots & a_{ij}B \end{array}\right]$$
z~cN(μ(α),ζ(α)) = the complex Gaussian distribution of the complex random vector *z* with mean μ and variance ζ, and α is a real-valued parameter vector that completely and uniquely specifies the distribution of z (see [42]).

## 2. Problem Formulation

### 2.1. Swarming UAV Synthetic Aperture

A swarming UAV synthetic aperture was presented in our early published paper [43]. Figure 1 shows a graphical representation of a UAV swarm as it morphs in time (iteration *I* in this paper). Each of the *M* agents in the swarm has a location, orientation, and trajectory. Notionally, these have position $P_{m,i}\left(r,\theta ,\phi \right)$, where *m* is the agent’s index and *i* is the iteration. During swarming, the agents undergo rotations and translations, where a dual quaternion framework provides a convenient mechanism to handle this behavior. This motion rotates the agents’ local *(u, v, w)* coordinate systems that describe the spatial orientation of their antenna radiation pattern with respect to (w.r.t.) the global coordinate system and the direction to the incoming signal of interest $S\left(\theta_{n},\phi_{n}\right)$, which is the *n*th source. The collection of these measurements over iteration creates a synthetic aperture that can be used to calculate the parameters of interest $\left(\theta_{n},\phi_{n}\right)$. Notionally, *K* is independent data sampled for each agent in each iteration and *K* is usually called a “snapshot”. 

### 2.2. Signal Model

Friedlander and Weiss presented a mutual coupling model in the presence of sensor mutual coupling, gain, and phase uncertainties [44]. We neglect the mutual coupling effect in the signal model since the spacing of aperiodic array reconstructed from swarming UAVs is expected to be much larger than a wavelength. Consider that an arbitrary array of *M* elements receive *N* uncorrelated incident signals in the far-field demonstrated in part 1. Thus, the received signal for the *i-*th iteration can be represented as
(2)$$X_{i}(k)=\Gamma_{i}\cdot {\tilde{Α}}_{i}\cdot S_{i}(k)+W_{i}(k)\;\;\;\;k=1,2,\cdots ,K;\;\;\;\;i=1,2,\cdots ,I$$
where $X_{i}(k)={\left[X_{1,1}(k),\cdots ,\;\;X_{1,M}(k),\cdots ,X_{i,1}(k),\;\;\cdots ,\;\;X_{i,M}(k)\right]}^{T}$, $S_{i}(k)={\left[S_{1,1}(k),\;\cdots ,\;\;S_{1,N}(k),\cdots ,S_{i,1}(k),\;\;\cdots ,\;\;S_{i,N}(k)\right]}^{T}$,
$W_{i}(k)={\left[W_{1,1}(k),\;\;\cdots ,\;W_{1,M}(k),\cdots ,W_{i,1}(k),\cdots ,\;W_{i,M}(k)\right]}^{T}$, $\Gamma_{i}=diag\{g_{1,1}e^{-j\omega_{0}\psi_{1,1}},\cdots ,\;\;g_{1,M}e^{-j\omega_{0}\psi_{1,M}},\cdots ,\;\;g_{i,1}e^{-j\omega_{0}\psi_{i,1}},\;\;\cdots ,\;\;g_{i,M}e^{-j\omega_{0}\psi_{i,M}}\}$,
${\tilde{A}}_{i,mn}=e^{-j\omega_{0}\tau_{i,mn}}$ and m=1,2,⋯,  M;  n=1,2,⋯,  N;i=1,2,⋯, I. Then,
(3)$$X(k)=\sum_{i=1}^{I}X_{i}(k)\;\;=\sum_{i=1}^{I}\left[\Gamma_{i}\cdot {\tilde{Α}}_{i}\cdot S_{i}(k)+W_{i}(k)\right]$$
therefore, $S(k)=\sum_{i=1}^{I}S_{i}(k)$,$W(k)=\sum_{i=1}^{I}W_{i}(k)$,$\Gamma =\sum_{i=1}^{I}\Gamma_{i}\;$, and $\tilde{A}=\sum_{i=1}^{I}{\tilde{A}}_{i}$. Note that the sensor gain $g_{i,m}$ and phase $\psi_{i,m}$ change w.r.t. element location based on orientation of UAV; $\tau_{i,mn}$ changes w.r.t. location; $S_{i}$ is constant; $W_{i}$ may change w.r.t. location, velocity of UAV, and environment.

Since the sources we consider here are in the far field from the observing array. It is easy to find that $\tau_{i,mn}$ can be represented by $\tau_{i,mn}\;=-d_{i,mn}/c$, and then
(4)$$d_{i,mn}=x_{i,m}\sin\theta_{n}\cos\phi_{n}+y_{i,m}\sin\theta_{n}\sin\phi_{n}+z_{i,m}\cos\theta_{n}$$
where $d_{i,mn}$ is the distance from origin (reference sensor) of the coordinate to the *m-*th sensor in the direction of the *n*th source for the *i-*th iteration, c is the propagating velocity in free space, $\left(x_{i,m},y_{i,m},z_{i,m}\right)$ are the coordinates of the *m-*th sensor for the *i-*th iteration, $\left(\theta_{n},\phi_{n}\right)$ are the DOAs of the *n-*th source in the sphere coordinate. From Equations (5), the matrix $\tilde{A}$ can be obtained by
(5)$${\tilde{A}}_{i,mn}\;=e^{j(\omega_{0}/c)(x_{i,m}\sin\theta_{n}\cos\phi_{n}+y_{i,m}\sin\theta_{n}\sin\phi_{n}+z_{i,m}\cos\theta_{n})}=e^{j(2\pi /\lambda )(x_{i,m}\sin\theta_{n}\cos\phi_{n}+y_{i,m}\sin\theta_{n}\sin\phi_{n}+z_{i,m}\cos\theta_{n})}$$
where λ is wavelength.

## 3. The CRB

### 3.1. Swarming UAV Synthetic Aperture

In theory, for a static array, the steering vector $\tilde{A}$ is considered invariant over different snapshots since the array geometry is invariant. Assume z~cN(μ(α),ζ(α)), then, the *m*,*n-*th general formula of the Fisher information matrix (FIM) on the covariance matrix of any unbiased estimate of α is:(6)Fmn=trζ−1(α)∂ζ(α)∂αmζ−1(α)∂ζ(α)∂αn  +2Re∂μ(α)∂αmζ−1(α)∂μ(α)∂αn
where $\alpha_{m}$ denotes the *m-*th component of α. The general formula has been presented in [45] and proved in [46]. 

Petre and Nehorai presented the deterministic and stochastic CRB in [46]. For deterministic CRB, the parameters, mean, and variance of the complex distribution are given by α=θ,Rest,Imstk=1K,σ2, $\mu (\alpha )={\left[As\left(k\right)\right]}_{k=1}^{K}$, $\zeta (\alpha )=block-diag\left(\sigma^{2}I\right)$. Then, the *m,n*-th FIM is
(7)$$F_{mn}=K\frac{\mu }{\sigma^{4}}\frac{\partial \sigma^{2}}{\partial \alpha_{m}}\frac{\partial \sigma^{2}}{\partial \alpha_{n}}+\frac{2}{\sigma^{2}}\;\;\cdot \sum_{k=1}^{K}{\left[\frac{\partial }{\partial \alpha_{m}}As\left(k\right)\right]}^{H}\left[\frac{\partial }{\partial \alpha_{n}}As\left(k\right)\right]$$

For stochastic CRB, the parameters, mean and variance of the complex distribution are given by α=θ,RePmn,ImPmnm,n=1K,σ2,$\mu \left(\alpha \right)=0$,$\zeta (\alpha )=block-diag\left(R\right)$. Then, the *m,n*-th FIM is
(8)$$F_{mn}=K\;\;tr\left[R^{-1}(\alpha )\frac{\partial R(\alpha )}{\partial \alpha_{m}}R^{-1}(\alpha )\frac{\partial R(\alpha )}{\partial \alpha_{n}}\right]$$

Since the signals estimated cannot be known completely and the signals in the practical are stochastic, this paper considers the stochastic CRB. 

### 3.2. CRB for the UAV Swarming System 

Before deriving the CRB, we assume that both incident signals and noise are stationary and the ergodic complex Gaussian random process with zero mean and nonsingular covariance matrix is uncorrelated with each other. The columns of $A=\Gamma \tilde{A}$ are linearly independent. An additional assumption is that the number of array elements reconstructed from swarming UAVs is greater than the number of sources. Therefore, the matrix of the steering vector has a full column rank.

The covariance matrices of signal, noise and observation vectors for *i-*th iteration are given by
(9)$$\begin{array}{c} P_{i}=E\left[S_{i}S_{i}^{H}\right],\sigma_{i}^{2}I_{0}=E\left[W_{i}W_{i}^{H},\right]\\ R_{i}\;=\;E\left[X_{i}(k)X_{i}{}^{H}(k)\right]=\Gamma_{i}{\tilde{A}}_{i}P_{i}{\tilde{A}}_{i}^{H}\Gamma_{i}^{H}+\sigma_{i}{}^{2}I_{0}\;=A_{i}P_{i}A_{i}^{H}+\sigma_{i}{}^{2}I_{0} \end{array}$$
where ${\tilde{A}}_{i}$ is the steering vector for the *i*-th iteration and $A_{i}≜\Gamma_{i}{\tilde{A}}_{i}$. It is useful to observe that if we let the sample data covariance matrix ${\hat{R}}_{i}=\;\frac{1}{K}\sum_{k=1}^{K}X_{i}(k)X_{i}{}^{H}(k)$, and $\hat{R}=\frac{1}{I}\sum_{i=1}^{I}{\hat{R}}_{i}$, then $\underset{K\to \infty }{\lim}{\hat{R}}_{i}=R_{i}$, and $\underset{I\to \infty }{\lim}\hat{R}=R$.

The log-likelihood function of *K* independent samples in *i-*th iteration of a zero-mean complex Gaussian random process $X_{i}\left(k\right)$ whose statistics depend on a parameter vector α is given by
(10)$$\begin{array}{ll} L_{i}\left(\alpha \right) & =-K\ln\left[\det\left(R_{i}\pi \right)\right]-\sum_{k=1}^{K}x_{i}{}^{H}\left(k\right)R_{i}{}^{-1}x_{i}\left(k\right)\\ & =Z-K\ln\left[\det\left(R_{i}\right)\right]-\sum_{k=1}^{K}x_{i}{}^{H}\left(k\right)R_{i}{}^{-1}x_{i}\left(k\right) \end{array}$$
where *Z* denotes the constant term of the log-likelihood function, det(R) represents the determinant of the matrix R, and $R_{i}$ is the time-varying covariance matrix w.r.t. the iteration. Thus, the log-likelihood function of X(k) is:(11)$$\begin{array}{ll} L\left(\alpha \right) & =-K\sum_{i=1}^{I}\ln[\det(R_{i}\pi )]-\sum_{i=1}^{I}\sum_{k=1}^{K}x_{i}{}^{H}(k)R_{i}{}^{-1}x_{i}(k)\\ & =\sum_{i=1}^{I}\left\{Z-K\ln[\det(R_{i})\right]-\sum_{k=1}^{K}x_{i}{}^{H}(k)R_{i}{}^{-1}x_{i}(k)\}\;=\sum_{i=1}^{I}L_{i}\left(\alpha \right) \end{array}$$

Therefore, the *m*,*n-*th elements of FIM are given by

(12)$$F_{mn}=-E\left[\frac{\partial^{2}L(\alpha )}{\partial \alpha_{m}\partial \alpha_{n}}\right]=-\sum_{i=1}^{I}E\left[\frac{\partial^{2}L_{i}(\alpha )}{\partial \alpha_{m}\partial \alpha_{n}}\right]\;\;\;=\sum_{i=1}^{I}\left\{K\;\;tr\left[R_{i}{}^{-1}(\alpha )\frac{\partial R_{i}(\alpha )}{\partial \alpha_{m}}R_{i}{}^{-1}(\alpha )\frac{\partial R_{i}(\alpha )}{\partial \alpha_{n}}\right]\right\}\;\;\;=\sum_{i=1}^{I}F_{i,mn}$$

It follows that the FIM’s submatrix $F_{mn}$ for the UAV swarming data collecting system can be obtained by summing the single-iteration $F_{i,mn}$ of FIM over the iterations. Furthermore, the FIM’s submatrix $F_{i,mn}$ can be obtained by multiplying the single-snapshot $F_{i,mn}$ and the number of snapshots. Thus, we only need to know the single-snapshot $F_{i,mn}$ for the *i-*th iteration. The problem of the major interest is the estimation of the incident angles of the sources. Expression of CRB for 1D DOA of each iteration for the present problem is listed in [47]. The CRB of 2D DOA estimation with arbitrary array for the *i-*th iteration, which can be considered as an arbitrary static array, presented in this paper is given in the Appendix A. 

## 4. Analysis of Single-Emitter Case

In this section, we investigate more details of the MUSB data collecting system with single-emitter case. The unknown parameters we consider here are the 2D DOAs $\left(\theta ,\phi \right)$. Assume the source variance is *P*, the noise variance is $\sigma^{2}$, the snapshot for each iteration is *K*, and the iteration is *I*. From the Appendix A, we have the formula of FIM w.r.t. 2D DOAs in the *i-*th iteration of the MUSB system. 

(13)Fi,s,2= Fi,θθ Fi,θϕFi,ϕθ  Fi,ϕϕ=2Kσ2ReDiHAi⊥Di⊙1212T⊗Ui

The *m*th element of the steering vector for the *i-*th iteration is given by
(14)$$a_{i,m}\left(\theta ,\phi \right)=\gamma_{i,m}\cdot \exp[j\frac{2\pi }{\lambda }(x_{i,m}\sin\theta \cos\phi +y_{i,m}\sin\theta \sin\phi +z_{i,m}\cos\theta )]$$
where $\gamma_{i,m}$ is the gain and phase parameter and can be represented as $\gamma_{i,m}=g_{i,m}e^{-j\omega_{0}\psi_{i,m}}$. Taking the derivative of $a_{i}\left(\theta ,\phi \right)$ w.r.t. θ and ϕ  for the *i-*th iteration, we obtain
(15)$$d_{i,\theta }=\frac{d}{d\theta }a_{i}\left(\theta ,\phi \right)=j\frac{2\pi }{\lambda }b_{i,m}⊙a_{i}\left(\theta ,\phi \right),\;d_{i,\phi }=\frac{d}{d\phi }a_{i}\left(\theta ,\phi \right)=j\frac{2\pi }{\lambda }q_{i,m}⊙a_{i}\left(\theta ,\phi \right)$$
where $b_{i,m}=x_{i,m}\cos\theta \cos\phi \;+y_{i,m}\cos\theta \sin\phi -z_{i,m}\sin\theta$,$q_{i,m}=-x_{i,m}\sin\theta \sin\phi +y_{i,m}\sin\theta \cos\phi$. Note

(16)$$A_{i}^{H}A_{i}=\sum_{m=1}^{M}g_{i,m}^{2}=G_{i,M}$$

Therefore, it is straightforward to verify that
(17)$$D_{i}^{H}\left(\theta \right)D_{i}\left(\theta \right)=\left(-j\frac{2\pi }{\lambda }b_{i}A_{i}^{H}\right)\left(j\frac{2\pi }{\lambda }b_{i}A_{i}\right)=\frac{4\pi^{2}}{\lambda^{2}}\sum_{m=1}^{M}b_{i,m}^{2}g_{i,m}^{2}$$
(18)$$D_{i}^{H}\left(\phi \right)D_{i}\left(\phi \right)=\left(-j\frac{2\pi }{\lambda }q_{i}A_{i}^{H}\right)\left(j\frac{2\pi }{\lambda }q_{i}A_{i}\right)\;\;=\frac{4\pi^{2}}{\lambda^{2}}\sum_{m=1}^{M}q_{i,m}^{2}g_{i,m}^{2}$$
(19)$$D_{i}^{H}\left(\theta \right)D_{i}\left(\phi \right)=\left(-j\frac{2\pi }{\lambda }b_{i}A_{i}^{H}\right)\left(j\frac{2\pi }{\lambda }q_{i}A_{i}\right)\;\;=\frac{4\pi^{2}}{\lambda^{2}}\sum_{m=1}^{M}b_{i,m}q_{i,m}g_{i,m}^{2}=D_{i}^{H}\left(\phi \right)D_{i}\left(\theta \right)$$

Substituting Equations (14), (15), and (17), we can obtain 
(20)$$\begin{array}{ll} D_{i}^{H}\left(\theta \right)A_{i}^{⊥}D_{i}\left(\theta \right) & =D_{i}^{H}\left(\theta \right)\left(I_{0}-A_{i}{\left(A_{i}^{H}A_{i}\right)}^{-1}A_{i}{}^{H}\right)D_{i}\left(\theta \right)=D_{i}^{H}\left(\theta \right)D_{i}\left(\theta \right)-\frac{D_{i}^{H}\left(\theta \right)A_{i}A_{i}^{H}D_{i}\left(\theta \right)}{G_{i,M}}\\ & =\frac{4\pi^{2}}{\lambda^{2}}\sum_{m=1}^{M}b_{i,m}^{2}g_{i,m}^{2}-\frac{4\pi^{2}}{\lambda^{2}}\frac{1}{G_{i,M}}{\left(\sum_{m=1}^{M}b_{i,m}g_{i,m}^{2}\right)}^{2}=\frac{4\pi^{2}}{\lambda^{2}}\{\sum_{m=1}^{M}b_{i,m}^{2}g_{i,m}^{2}-\frac{1}{G_{i,M}}{\left(\sum_{m=1}^{M}b_{i,m}g_{i,m}^{2}\right)}^{2}\} \end{array}$$

Using the same derivative procedure, we can obtain
(21)$$D_{i}^{H}\left(\phi \right)A_{i}^{⊥}D_{i}\left(\phi \right)=\;\frac{4\pi^{2}}{\lambda^{2}}\left\{\sum_{m=1}^{M}q_{i,m}^{2}g_{i,m}^{2}-\frac{1}{G_{i,M}}\right.\cdot \left.{\left(\sum_{m=1}^{M}q_{i,m}g_{i,m}^{2}\right)}^{2}\right\}$$
(22)$$D_{i}^{H}\left(\theta \right)A_{i}^{⊥}D_{i}\left(\phi \right)=\;\frac{4\pi^{2}}{\lambda }\left\{\sum_{m=1}^{M}b_{i,m}q_{i,m}g_{i,m}^{2}-\frac{1}{G_{i,M}}\right.\;\left.\;\cdot \left(\sum_{m=1}^{M}b_{i,m}g_{i,m}^{2}\sum_{m=1}^{M}q_{i,m}g_{i,m}^{2}\right)\right\}=D_{i}^{H}\left(\theta \right)A_{i}^{⊥}D_{i}\left(\phi \right)$$
Furthermore,
(23)$$U_{i}=PA_{i}^{H}R_{i}^{-1}A_{i}P=P^{2}A_{i}^{H}{\left(A_{i}^{H}PA_{i}+\sigma^{2}I_{0}\right)}^{-1}A_{i}=P^{2}G_{i,M}{\left(PG_{i,M}+\sigma^{2}\right)}^{-1}=\frac{P^{2}G_{i,M}}{PG_{i,M}+\sigma^{2}}$$
Therefore,
(24)$$F_{i,\theta \theta }=\frac{8K\pi^{2}}{\lambda^{2}\sigma^{2}}\frac{P^{2}G_{i,M}^{2}}{PG_{i,M}+\sigma^{2}}\left\{\frac{1}{G_{i,M}}\sum_{m=1}^{M}b_{i,m}^{2}g_{i,m}^{2}-{\left(\frac{1}{G_{i,M}}\sum_{m=1}^{M}b_{i,m}g_{i,m}^{2}\right)}^{2}\right\}$$
(25)$$F_{i,\phi \phi }=\frac{8K\pi^{2}}{\lambda^{2}\sigma^{2}}\frac{P^{2}G_{i,M}^{2}}{PG_{i,M}+\sigma^{2}}\left\{\frac{1}{G_{i,M}}\sum_{m=1}^{M}q_{i,m}^{2}g_{i,m}^{2}-{\left(\frac{1}{G_{i,M}}\sum_{m=1}^{M}q_{i,m}g_{i,m}^{2}\right)}^{2}\right\}$$
(26)$$F_{i,\theta \phi }=\frac{8K\pi^{2}}{\lambda^{2}\sigma^{2}}\frac{P^{2}G_{i,M}^{2}}{PG_{i,M}+\sigma^{2}}\left\{\frac{1}{G_{i,M}}\sum_{m=1}^{M}b_{i,m}q_{i,m}g_{i,m}^{2}-\left(\frac{1}{G_{i,M}^{2}}\sum_{m=1}^{M}b_{i,m}g_{i,m}^{2}\sum_{m=1}^{M}q_{i,m}g_{i,m}^{2}\right)\right\}=F_{i,\phi \theta }$$
Assume
(27)$$B_{i}=\frac{1}{G_{i,M}}\sum_{m=1}^{M}b_{i,m}^{2}g_{i,m}^{2}-{\left(\frac{1}{G_{i,M}}\sum_{m=1}^{M}b_{i,m}g_{i,m}^{2}\right)}^{2},\;Q_{i}=\frac{1}{G_{i,M}}\sum_{m=1}^{M}q_{i,m}^{2}g_{i,m}^{2}-{\left(\frac{1}{G_{i,M}}\sum_{m=1}^{M}q_{i,m}g_{i,m}^{2}\right)}^{2}$$
(28)$$V_{i}=\frac{1}{G_{i,M}}\sum_{m=1}^{M}b_{i,m}q_{i,m}g_{i,m}^{2}-\left(\frac{1}{G_{i,M}^{2}}\sum_{m=1}^{M}b_{i,m}g_{i,m}^{2}\sum_{m=1}^{M}q_{i,m}g_{i,m}^{2}\right),\;C_{i}=\frac{G_{i,M}^{2}SNR^{2}}{G_{i,M}SNR+1}$$
where $SNR=P/\sigma^{2}$. Thus, summing over iteration, we obtain the CRB
(29)$$CRB={\left[\begin{array}{cc} F_{\theta \theta } & F_{\theta \phi }\\ F_{\phi \theta } & F_{\phi \phi } \end{array}\right]}^{-1}$$
where $F_{\theta \theta }=\frac{8K\pi^{2}}{\lambda^{2}}\sum_{i=1}^{I}C_{i}B_{i}$, $F_{\phi \phi }=\frac{8K\pi^{2}}{\lambda^{2}}\sum_{i=1}^{I}C_{i}Q_{i}$, $F_{\theta \phi }=\frac{8K\pi^{2}}{\lambda^{2}}\sum_{i=1}^{I}C_{i}V_{i}=F_{\phi \theta }$. If we ignore the sensor orientation (i.e., Γ=1) and let *K* = 1, then $A_{i}^{H}A_{i}=M$ and the FIM w.r.t. 1-D DOA is given by
(30)$$F_{\theta }=\frac{8\pi^{2}}{\lambda^{2}}\frac{M^{2}SNR^{2}}{MSNR+1}B$$
where $B=\sum_{i=1}^{I}B_{i}$ and $B_{i}=\frac{1}{M}\sum_{m=1}^{M}b_{i,m}^{2}-{\left(\frac{1}{M}\sum_{m=1}^{M}b_{i,m}\right)}^{2}$, which coincides with the results in [47].

## 5. The Algorithm

### 5.1. UAV Parameters

Assume there are *M* swarming UAVs and each UAV swarms in a cylinder region (radius *r*, *h* = 2r). Each UAV has an initial location (x, y, z) in the swarming region with a vector velocity $\vec{V_{i,m}}$ in each iteration and the initial locations of the *M* UAVs are considered as the first iteration. The swarming short distance in each iteration is represented as a vector $\vec{d_{i,m}}$. The scalar quantity can be represented as *d*, which uses the following relation:(31)$$d=\beta_{t}\cdot \lambda /\rho$$
where $\beta_{t}$ is a uniformly distributed random number matrix between zero and one, *λ* is wavelength and *p* is the coefficient determining the distribution mean of swarm distance. Figure 2 shows the distribution of short distance with mean μ = 0.25 wavelength. Here, the range of *d* also depends on the speed of UAV and data sampling interval and can be configured by the customer. We can obtain

(32)$$\vec{d_{i,m}}=\vec{V_{i,m}}\cdot t$$

### 5.2. Data Processing and Algorithm

When UAV swarms, a significant amount of data/information may be sampled due to the number of swarming UAVs and number of iterations. Herein, the location of each UAV is considered as one data point. Thus, when *M* UAVs morph *I* times, we have M∗I data points which will be used to reconstruct virtual (e.g., synthetic) 3D aperiodic arrays to calculate the MUSIC spectrum and estimate DOAs. The number of data points (represented as $N_{d}$) is equivalent to the number of elements for a static array. 

One challenge is how to choose *N_d_* for computing the MUSIC spectrum at each signal processing iteration (represented as p-iteration). We would like to use more $N_{d}$ in each p-iteration since more $N_{d}$ used for MUSIC spectrum calculation each time means that more array elements are used for data processing in a static antenna array, and are more accurate for DOA estimation. However, more data processing points cause higher calculation cost. Thus, it is necessary to compromise *N_d_* at each *p*-iteration and computational complexity. 

Another problem is how to well choose data collecting and processing methods. Wide range of methods can be deployed to collect, concatenate, and process data collected from time-dependent measurements. MUSIC algorithm requires a minimum of three unique samples collected from these time-dependent measurements to provide DOA estimation. The required data samples can be obtained from agents in the swarm (e.g., one UAV locally collecting three samples at different times, or three spatially distributed UAVs collecting one sample simultaneously). We need to consider spatiotemporal distribution of samples (such as sampling rate w.r.t. wavelength, trajectory and velocity of UAVs, orientation of sensors, etc.) and iterative processing of measurements (use all data collected, truncate or applying a moving window, etc.). The iterative processing method with a moving window is called the iterative-MUSIC used in this paper.

Figure 3 shows the data processing schematic for three data points in each iteration within the MUSB system. The left part in Figure 3 shows a UAV swarm and sampling method, and the right part shows the algorithm and data processing method. When UAV swarms to a certain location, we will sample *K* times and each snapshot takes time $d_{t}$. After taking *K* snapshots, the program sets up a data point. When *N_d_* = 3, the program computes the stand MUSIC spectrum using three data points and stores the result. Then when UAV swarms again, we accumulate the current data point and two previous data points to calculate the MUSIC spectrum. Then we multiply the current and previous MUSIC spectrum at each p-iteration where we obtain the iterative-MUSIC spectrum to reduce noise level and improve DOA estimation performance. Note that if the p-iteration is too big, the value of spectral points will be very small and might be taken as zero. If so, DOA cannot be estimated and we may use dB instead of a number at that situation. Then, we refine and estimate DOAs and use predefined DOA estimation precision criteria to stop the process.

### 5.3. MUSIC Algorithm for MUSB Array

As stated in part 2 of this section, the number of reconstructed MUSB arrays is *N_d_*. Here, we rewrite the covariance of signals listed in Part 2 of Section 3 and give the first *p*-iteration (slightly different from UAV swarming iteration) of the iterative-MUSIC algorithm. 

Rewriting the data model (1), we obtain
(33)$$X(k)=\Gamma \cdot \tilde{Α}\cdot S(k)+W(k)\;\;\;=A\cdot S(k)+W(k)\;\;\;\;\;\;\;\;\;\;\;\;\;\;\;\;\;\;\;\;\;\;\;\;\;\;\;\;\;\;\;k=1,2,\cdots ,K$$
where $\left\{X\left(k\right)\in C^{N_{d}\times 1}\right\}$ are the vectors of sampled data, $\left\{S\left(k\right)\in C^{N\times 1}\right\}$ are the source signals, $\left\{\Gamma \in C^{N_{d}\times N_{d}}\right\}$ are the gain and phase of sensors and $\left\{\tilde{A}\in C^{N_{d}\times N}\right\}$ are the regular steering vectors. Thus, the covariance of $X\left(k\right)$ is
(34)$$R\;\;=\;\;E\left[X(k)X^{H}(k)\right]=\Gamma \tilde{A}P{\tilde{A}}^{H}\Gamma^{H}+\sigma^{2}I_{0}=APA^{H}+\sigma^{2}I_{0}$$
Define

(35)$$R_{s}=APA^{H}$$

$R_{s}$ is $N_{d}\times N_{d}$ matrix with rank *N*. Therefore, it has $N_{d}-N$ repeated eigenvectors corresponding to the minimum eigenvalues $\sigma^{2}$. Let $e_{i}$ be such an eigenvector so that $R_{s}e_{i}\;=\;0$ or $A^{H}e_{i}\;=\;0$, thus, $N_{d}-N$ eigenvectors $e_{i}$ corresponding to the minimal eigenvalues are orthogonal to each of *N* signal columns of $A=\Gamma \tilde{A}$, proved in [5]. *N_d_* − *N* dimensional subspace spanned by the noise eigenvectors is defined as noise subspace and *N* dimensional subspace spanned by incident signal mode vectors is defined as signal subspace.

Let $Q_{n}$ be $N_{d}\times \left(N_{d}-N\right)$ matrix of noise eigenvectors, then the MUSIC spatial spectrum function is given by
(36)$$P_{MU}(\theta ,\phi )=\;\frac{1}{\tilde{a}{\left(\theta ,\phi \right)}^{H}\Gamma^{H}Q_{n}Q_{n}{}^{H}\Gamma \tilde{a}(\theta ,\phi )}=\;\frac{1}{{\left\|Q_{n}{}^{H}\Gamma \tilde{a}(\theta ,\phi )\right\|}^{2}}=\frac{1}{{\left\|Q_{n}{}^{H}a(\theta ,\phi )\right\|}^{2}}$$

Then, the search spectrum peaks in the range of θ and ϕ, and the peak spectrum points we obtain are the estimation of arrival angles of incident waves. 

### 5.4. Convergence Check

The algorithm performs the calculation until the system converges. The convergence can be guaranteed since the estimated DOA is a convergent series. 

When the signal is covered by a high noise level, the estimated DOAs might be far from the ground truth and cannot be judged for convergence. However, as the iteration increases, the noise level is reduced and estimated DOAs are converged gradually. The Equation for judging convergence is given by
(37)$$DOA_{i+1}-DOA_{i}\leq \varepsilon ,\;\;\;\;\;\;\;\;\;\;\;\;\;\;\;\;\;\;\;\;\;\;\;\;\;\;\;i=1,2,\cdots ,I_{m}$$
where $I_{m}$ is the number of p-iteration of the iterative-MUSIC algorithm and ε is the preset threshold. The numerical simulation is given in part 1 of Section 6.

### 5.5. Computational Complexity Analysis

The orders of computational complexities of conventional spectral MUSIC [5], iterative-MUSIC, and eigenstructure-based algorithm with array interpolation (denoted by array interpolation) [48] are compared in Table 1. In this table, we assume the interpolated number of the array interpolation technique is equal to the number of actual sensors (*M_1_* = *N_d_* ) and the total number of angular sectors is denoted as $I_{\theta }I_{\phi }$. Those methods in Table 1 include the eigendecomposition step represented by the term $O\left(N_{d}^{2}N\right)$ and the computation of $J_{\theta }J_{\phi }$ samples of the MUSIC null-spectrum function represented by $O\left(J_{\theta }J_{\phi }\left(N_{d}+1\right)\left(N_{d}-N\right)\right)$, where $J_{\theta }$ and $J_{\phi }$ stand for the search numbers along the directions of θ and ϕ. ${\tilde{J}}_{\theta }{\tilde{J}}_{\phi }$ stands for the total search numbers for each sector of array interpolation algorithm. The iterative-MUSIC at each iteration may have very low snapshot (*K* = 1) different from the traditional spectral MUSIC algorithm which requires high snapshots. The computation complexity of the iterative-MUSIC algorithm is a little bit lower than the array interpolation at each iteration ($J_{\theta }J_{\phi }\approx I_{\theta }I_{\phi }{\tilde{J}}_{\theta }{\tilde{J}}_{\phi }>>N_{d}$), but it is higher in the whole UAV swarming period. However, the array interpolation algorithm will be very complex when applied for 3D random arrays. 

## 6. Simulation and Measurement Results

In this section, several groups of simulations will be carried out to demonstrate the performance of the presented distributed directional finding system in this paper. As the framework of the MUSB system established in this paper is mentioned for the first time, we focus mainly on analyzing the impact of various factors on the feasibility of the MUSB system. Practical measurements in the lab are also given in part 3 to show the DOA estimation performance in practice. 

The wavelength of signal is fixed at 1 meter (m) and the simulation in each scenario is repeated 500 times. The elevation angle of the source emitter is 85° and the azimuth angle is 270°. As one UAV swarms till *N_d_* = 3, the iterative-MUSIC algorithm begins to search (0°, 179°) space for elevation angle and (0°, 359°) space for azimuth angle with 1° interval to form the overcomplete MUSIC spectrum for each p-iteration. Then, UAV keeps swarming and the iterative-MUSIC algorithm keeps computing the MUSIC spectrum before the precision of DOA estimation is satisfied. When the preset threshold is satisfied, UAV stops swarming and the reconstructed process of phased arrays is terminated. The refined DOA estimations are obtained by scanning the reconstructed signal peaks from the iterative-MUSIC algorithm with 0.1° step during the refinement procedure 10 times. 

Moreover, the speed of UAV will influence the snapshot at each location where the system samples the source emitter. The snapshot at each location should be very low if the UAV swarms very fast and the snapshot can be high when the swarming rate of UAV is low. The speed of UAV in this paper will be represented by the distance between two iterations of swarming UAVs shown in Figure 2. 

The joint root-mean-square error (RMSE) of incident signals is used for statistical DOA estimation precision evaluation, which is defined as
(38)$$RMSE_{\theta ,\phi }=\sqrt{\sum_{w=1}^{W}\sum_{n=1}^{N}\left[{\left({\hat{\theta }}_{n}^{w}-\theta_{n}^{w}\right)}^{2}+{\left({\hat{\phi }}_{n}^{w}-\phi_{n}^{w}\right)}^{2}\right]/2WN}$$
where *W* is the number of Monte Carlo simulations, $\left(\theta_{n}^{w},\phi_{n}^{w}\right)$ represent the actual DOAs of the *n*th signal, and $\left({\hat{\theta }}_{n}^{w},{\hat{\phi }}_{n}^{w}\right)$ represent the estimated DOAs of the *n*th signal in the *w-*th simulation. 

First, the system convergence will be studied in a typical scenario. Second, the DOA estimation performance using the iterative-MUSIC algorithm will be compared with CRB in various scenarios. Finally, DOA estimation performance in practice will be investigated. 

### 6.1. System Convergence 

Figure 4 provides and example of the MUSB distributed directional finding system gradually converging to the ground truth as iteration increases by using iterative-MUSIC algorithm. 

### 6.2. DOA Estimation Performance 

The performance of DOA estimation depends on multiple factors such as SNR, *K*, *N_d_*, velocity of UAV and number of iterations. Furthermore, the performance also depends on the distinctness of the array geometries due to the diversity of different observations at different time instants. In this section, we only consider the single-emitter case. When a single source is present, a typical scenario is set in which UAV swarming short distance mean *µ* = 2*λ*, the Monte Carlo simulation number and angles of the source are the same as before. Figure 5 depicts the results predicted by the stochastic CRB derived in Section 4. The joint RMSE of elevation and azimuth angles obtained from the iterative-MUSIC algorithm together with CRB from Section 4 are shown in Figure 6 and Figure 7.

The extreme case is *K* = 1, *N_d_*= 3 (2D DOA estimation requires a minimum of three unique samples). The fixed settings and changed settings are listed as: Figure 5a varies snapshots *K* from 1 to 96 for different SNR with iteration *i* = 100, *N_d_*= 3; Figure 5b varies SNR from −30 to 0 dB for different UAV iterations with *K* = 1, *N_d_* = 3; Figure 5c varies SNR from -30 to 0 dB for different data points *N_d_* in each iteration with *K* = 1, *i* = 100; Figure 5d varies SNR for different speed of UAV with *K* = 1, *N_d_* = 3, *i* = 100; Figure 6a varies SNR of one signal from -20 to 20 dB with *K* = 1, *N_d_* = 3; Figure 6b varies *K* from 1 to 99 with SNR = 0 dB, *N_d_* = 3; Figure 6c varies speed of UAV with *K* = 1, *N_d_* = 3, SNR = 0 dB; Figure 6d varies incident elevation angles from 5° to 90° with SNR = 0 dB, *N_d_* = 3, *K* = 1; Figure 7a varies sensor gain deviation with SNR = 0 dB, *N_d_* = 3, *K* = 1; Figure 7b varies sensor phase deviation with SNR = 0 dB, *N_d_* = 3, *K* = 1. Scenarios (a)–(d) in Figure 5 give the lower bounds of the MUSB system. Scenarios (a)–(d) in Figure 6 show the performance of MUSB array without element rotation (sensor gain *g* = 1 and sensor phase *ψ* = 0 and scenarios (a)–(b) in Figure 7 show the impact factors with the element rotation of MUSB array (sensor gain and phase coefficients have certain deviations). 

From Figure 6 and Figure 7, we can find that Figure 6a shows the DOA estimation performance of the MUSB system will increase with increasing SNR; Figure 6b shows that the system can estimate DOAs even when snapshot *K* = 1; Figure 6c shows that when the UAV speed increases, the precision of DOA estimation increases; Figure 6d shows that the precision of DOA estimation increases with increasing elevation angles. Figure 7 shows the performance of the MUSB array with UAV rotating associated with sensor gain and phase varying. Figure 7a shows that the DOA estimation is not significantly impacted as the standard deviation of sensor gain increases; Figure 7b shows the precision of DOA estimation decreases significantly when the standard deviation of the sensor phase increases to a certain value. 

In Figure 7b, we find that the stochastic CRB is flat for the single-emitter case as the phase standard phase varies because Equation 16 shown in Section 4 cancels the sensor phase errors by multiplying the steering vector *A* and the conjugate transpose of *A*, while the iterative-MUSIC algorithm does not have the advantages CRB has utilized. Equation 39 shows the advantages of CRB for the single-emitter case and Equations 16–21 give the derivation process with advantages of sensor phase error cancel.
(39)$$A_{i}^{H}A_{i}=\left[g_{i,1}\cdot e^{-j\varphi_{1,1}},\cdots ,g_{i,m}\cdot e^{-j\varphi_{1,m}}\right]\left[\begin{array}{l} g_{i,1}\cdot e^{j\varphi_{1,1}}\\ \;\;\vdots \\ g_{i,m}\cdot e^{j\varphi_{1,m}} \end{array}\right]=\sum_{m=1}^{M}g_{i,m}^{2}$$
where $\varphi_{i,m}$ includes the sensor phase deviation and phase difference from sensor *m* to relative phase center in the *i-*th iteration. 

### 6.3. Measurement 

In the experiment, the test fixture provides a convenient platform to study this morphing in time (using sixteen elements). Randomly positioned rectangular patch antennas designed for 2.45 GHz are used with a randomly morphing volume provided by a moving platform named as “Medusa” to estimate DOAs. Figure 8 shows that one source in the far-field transmits and receiving sensors receive the signal with the vector network analyzer (VNA) measuring the *S_21_* of transmitting and receiving antennas. Figure 9 shows the measured MUSIC spectrum with 16 spatially distributed elements as a static volumetric random array in Medusa with element rotation (Rotate 0–45 degrees around x, y, z axis randomly, no UAV swarming). Figure 10 shows the measured azimuth and elevation errors gradually converge from around 10–20 degrees to around 1–2 degrees as the iteration increases with UAV swarming. In Figure 10, we take the extreme case ($N_{d}$ = 3, *K* = 1).

## 7. Conclusions

This paper establishes a MUSB data collection framework for the first time, which makes it possible to realize source estimation with low snapshot under low SNR environment in a UAV swarming period. Theoretical and experiment results are given to reveal the performance of the MUSB phased array system used for 2D DOA estimation, which supports the feasibility of the system. The iterative-MUSIC algorithm is applied for the framework and it can estimate the DOAs efficiently only with one snapshot in each iteration when UAV swarms very fast. The UAV speed controls the structure of the reconstructed aperiodic phased arrays from the MUSB system and the DOA estimation precision is increased when the distance between the two iterations of swarming UAV is increased. The impact of known sensor gain errors and phase errors from the UAV rotation for the DOA estimation performance are also investigated. Practical experiment results match the theoretical expectation of the MUSB system using the iterative-MUSIC algorithm. Our results will benefit future research on performance analysis and optimal design of time-varying antenna arrays based on the UAV swarm. It is also interesting to extend the results when position errors are present in the future. 

## Figures and Tables

**Figure 1 sensors-19-02659-f001:**
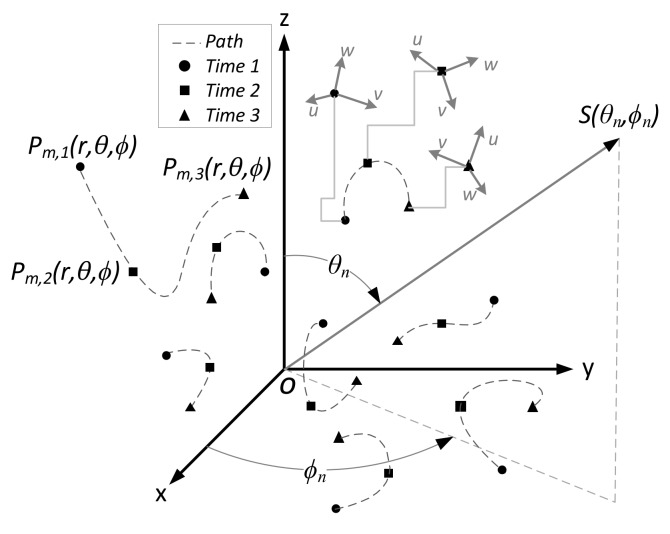
Morphing micro-UAV swarm-based (MUSB) antenna array configuration.

**Figure 2 sensors-19-02659-f002:**
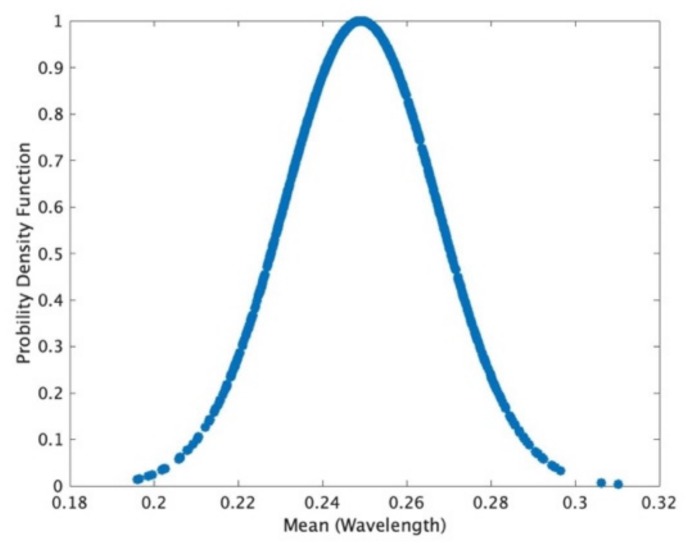
Distribution of short distance that one unmanned aerial vehicle (UAV) swarms.

**Figure 3 sensors-19-02659-f003:**
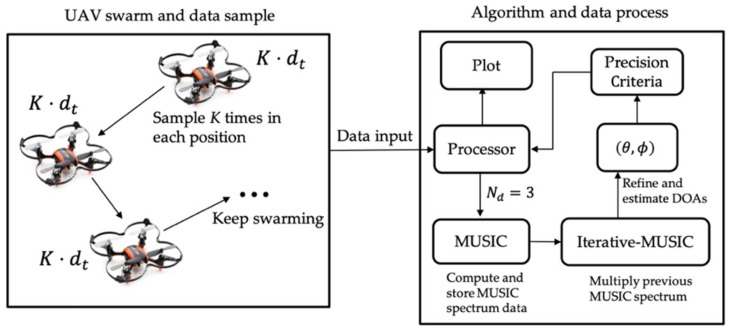
Data processing schematic for MUSB system.

**Figure 4 sensors-19-02659-f004:**
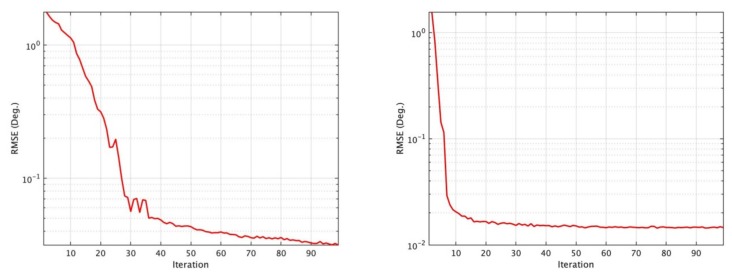
Direction-of-arrival (DOA) estimation convergence for the MUSB system. (**a**) *N_d_* = 3, SNR = 0 dB, *K* = 1. (**b**) *N_d_* = 3, SNR = 0 dB, *K* = 16.

**Figure 5 sensors-19-02659-f005:**
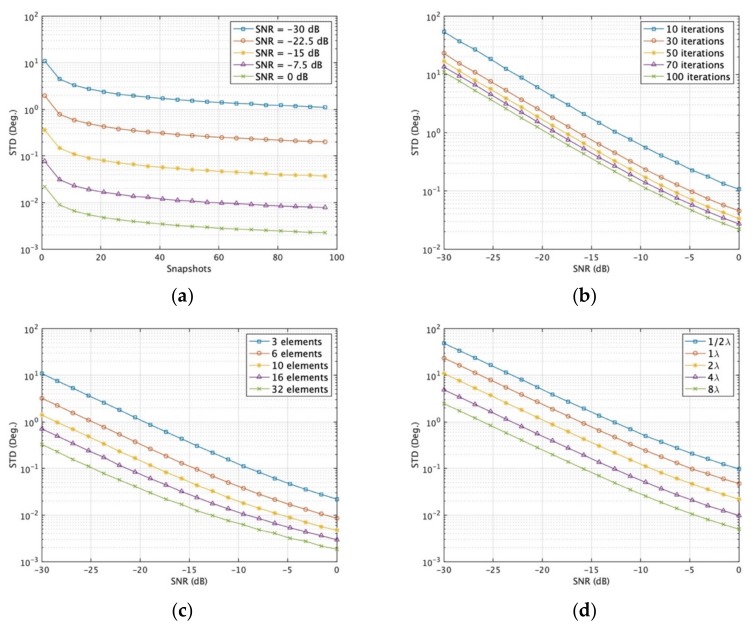
DOA standard deviation predicted by the stochastic Cramer–Rao bound (CRB). (**a**) Varying snapshots with different signal-to-noise ratio (SNR); (**b**) varying SNR with different iterations of UAV swarming; (**c**) varying SNR with different array element number; (**d**) varying SNR with different average speed.

**Figure 6 sensors-19-02659-f006:**
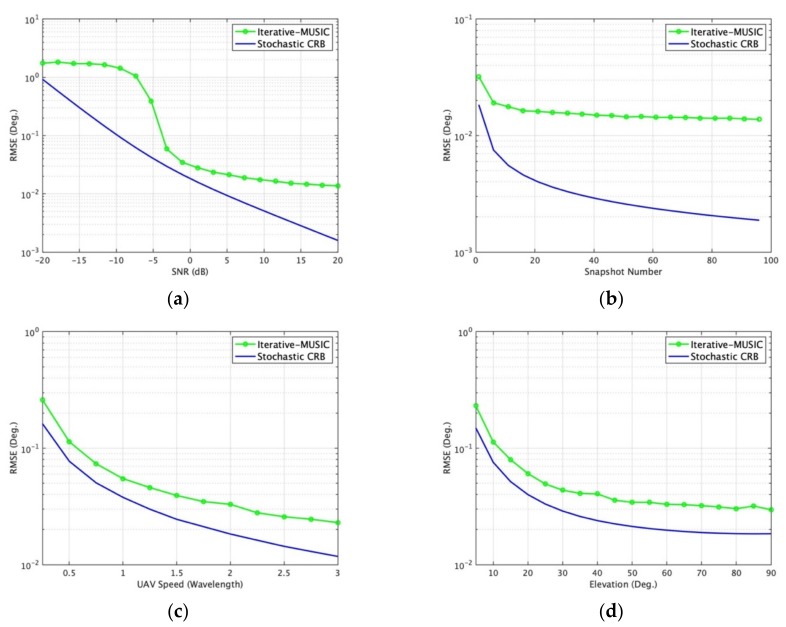
DOA estimation root mean square error (RMSE) of iterative-MUSIC (multiple signal classification) and CRB in different scenarios. (**a**) Varying SNR ($N_{d}$ = 3); (**b**) varying *K*; (**c**) varying the speed of UAV w.r.t. wavelength; (**d**) varying incident elevation angles.

**Figure 7 sensors-19-02659-f007:**
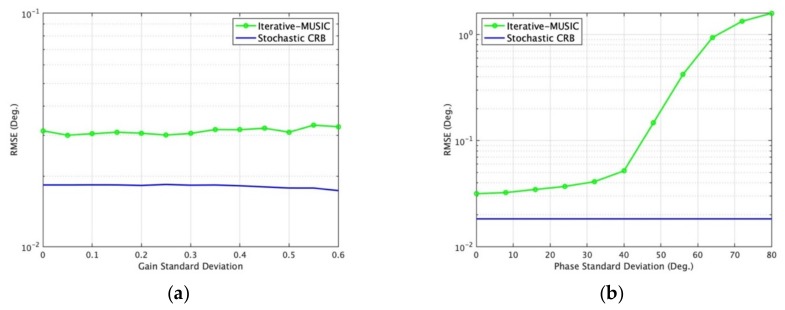
DOA estimation RMSE of iterative-MUSIC and CRB in different scenarios. (**a**) Varying sensor gain standard deviation; (**b**) varying sensor phase standard deviation.

**Figure 8 sensors-19-02659-f008:**
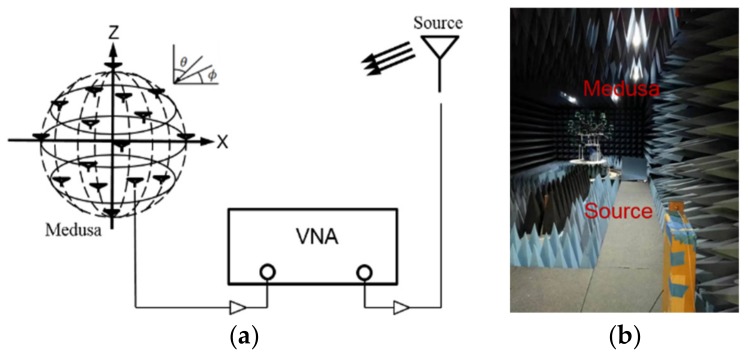
Test diagram. (**a**) Schematic diagram; (**b**) practical test diagram.

**Figure 9 sensors-19-02659-f009:**
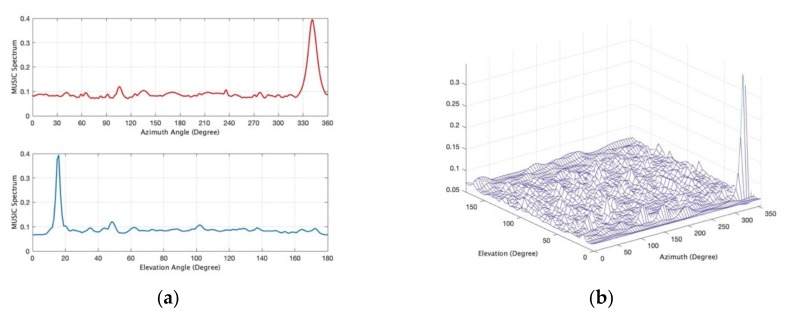
Measured MUSIC spectrum with iteration *i* = 1, *K* = 1, $N_{d}$ = 16, an incident signal of azimuth 356.3° and elevation 18°. (**a**) 2D MUSIC spectrum; (**b**) 3D MUSIC spectrum.

**Figure 10 sensors-19-02659-f010:**
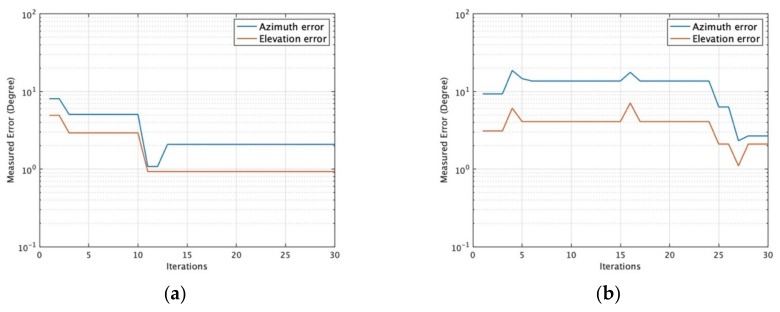
Measured errors compared to iterations with $N_{d}$ = 3, *K* = 1. (**a**) Element without rotation; (**b**) element with rotation (element rotates 0–45 degrees around x, y, z axis randomly to simulate UAV flying).

**Table 1 sensors-19-02659-t001:** The orders of computational complexities of real-valued operations.

Algorithm	Primary Computations
Spectral MUSIC	$O\left(N_{d}^{2}N+J_{\theta }J_{\phi }\left(N_{d}+1\right)\left(N_{d}-N\right)\right)$
Iterative-MUSIC	$O\left(N_{d}^{2}N+J_{\theta }J_{\phi }\left(N_{d}+1\right)\left(N_{d}-N\right)\right)$ for each iteration
Array Interpolation	$O\left(I_{\theta }I_{\phi }\left(N_{d}^{2}N+{\tilde{J}}_{\theta }{\tilde{J}}_{\phi }\left(N_{d}+1\right)\left(N_{d}-N\right)\right)\right)$

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
