# Peer review of "A Sensor-Driven Analysis of Distributed Direction Finding Systems Based on UAV Swarms"

_sensors, 2019, doi:10.3390/s19122659_

Reviewer 1 Report

The manuscript reported a micro-UAV swarm-based (MUSB) aperiodic antenna arrays for direction-of-arrival (DOA) estimation, it is interesting. Comments as follows: 1. The computation complexity analysis of each iteration is needed. 2. The real-time characteristic of the estimation framework, and it's relationship with UAV's speed and numbers could be discussed.

Author Response

The manuscript reported a micro-UAV swarm-based (MUSB) aperiodic antenna arrays for direction-of-arrival (DOA) estimation, it is interesting.

 Point 1: The computation complexity analysis of each iteration is needed. 

 Response 1: We add computational complexity analysis in section 5.

 Point 2: The real-time characteristic of the estimation framework, and it's relationship with UAV's speed and numbers could be discussed.

 Response 2: We have talked about the real-time characteristic of the estimation framework in section 5 and 6. The UAV speed will impact the sample rate and the number of snapshot in each iteration. When the UAV speed is high, the snapshot should be low or even equal to one; when the speed of UAV is low, the snapshot can be high. How good the real-time characteristic of the estimation framework depends on many impact factors, such as algorithm, hardware computing performance (FPGA, DSP, est.), UAV speed, sample rate, etc.  The snapshot of most of the results I gave is one. I used the swarming short distance with respect to wavelength to represent the UAV speed and the simulation results are shown in Figure 5 and 6. Furthermore, we have discussed the number of data points and number of UAVs. The number of data points is similar to the number of elements for normal static arrays. The number of UAVs will impact the first calculation and estimation of source because if the number of UAVs is 10, we only need one iteration to construct 10-element array. But if we only have 1 UAV, we need 9 iterations to construct 10-element array. So, the number of data points is much more important than the number of UAVs. The chosen of UAV number and number of data points depend on the data processing methods which are discussed in section 5.

Reviewer 2 Report

The authors have proposed an algorithm to measure the accuracy and efficiency of swarming UAV-based radio frequency (RF) and microwave data collection system. The paper is in good shape. However,

1> the authors should compare their algorithm with state-of-the-art and provided data numerically. 2> the authors mentioned that swarm formation is arbitrary but known. This is self-contradictory. If it is known it cannot be arbitrery and vice-versa.

Author Response

The authors have proposed an algorithm to measure the accuracy and efficiency of swarming UAV-based radio frequency (RF) and microwave data collection system. The paper is in good shape. However,

 Point 1: The authors should compare their algorithm with state-of-the-art and provided data numerically.

 Response 1: The goals of our paper are to introduce the micro-UAV swarm-based (MUSB) antenna array system used for 2-dimensional DOA estimation with low snapshot under low SNR environment and to investigate the fundamental limitations (error bounds) on DOA estimation imposed by the physical RF sensing mechanism of the MUSB system. The iterative-MUSIC algorithm we presented is not the main part of our paper. Therefore, we derived the Cramer-Rao bound (CRB) with gain and phase coefficient for our MUSB system and analysed the performance of the system based on the CRB. Furthermore, there are very limited algorithms which can be applied for 3-dimensional random time-varying arrays in low snapshot low SNR application environment. We know that some of machine learning algorithms attract more and more attentions and we may research for more algorithms to our MUSB system in the future.

 Point 2: The authors mentioned that swarm formation is arbitrary but known. This is self-contradictory. If it is known it cannot be arbitrary and vice-versa.

 Response 2: The “swarm formation is arbitrary but known” means the UAV swarming trajectory is random but the position at each sampling position is known (i.e. no position errors of sensors).  We already deleted this sentence. 

Reviewer 3 Report

The manuscript titled “A Sensor-Driven Analysis of Distributed Direction Finding Systems Based on UAV Swarms” represents well written, original scientific research. This paper presents a novel investigation of factors that impact the accuracy and efficiency of an UAV based on the radio frequency and microwave data collection. The idea of the research is interesting and present enough novelty. The paper title is accurate and concise. The abstract is well written and concise. In the entire manuscript, authors use a standard technical and scientific terminology. The manuscript is very well written and structured. After well written and concise Introduction, authors explained in detailed research problem, methodology and their novel algorithm. The simulation and measurement results were conducted according to the scientifically correct approach. The manuscript topics fit in Sensors journal scope, especially in Sensor technology and applications, remote sensors, as well as, signal processing. I recommended this paper to be accepted after minor revisions.

Comments for authors:

1.      The paper has some typos and language errors (e.g. line 90 – “Corner et al.” -> “Corner and Lamont”; line 125 – add equation number and renumber equation numbering in entire manuscript; line 129 – renumber reference [46] base on the Instructions for authors; line 147 – “Friedlander et al.” -> “Friedlander and Weiss”; in entire manuscript change “nth” to “n-th”; line 447 – “7 (b) -> “7b”; etc.).

2.      Variable name “const” is not adequate for the variable. Please use one letter for a variable name, accordingly to the “Instructions for authors”.

3.      The variable names must have same font style and size in equations, on figures and in the manuscript text. Please describe/introduce all variables used in equations or on figures in the manuscript text.

4.      All known equations must be adequately cited in the entire paper.

5.      Use MDPI standard font (Palatino Linotype) on figures if you can.

6.      Please introduce abbreviations if you want to use it abstract or in the manuscript text (e.g. DOA, CRB, FIM etc.).

7.      Please, double check all references and reference style.

8.      I suggest to slightly expand the Conclusions with main results.

Author Response

The manuscript titled “A Sensor-Driven Analysis of Distributed Direction Finding Systems Based on UAV Swarms” represents well written, original scientific research. This paper presents a novel investigation of factors that impact the accuracy and efficiency of an UAV based on the radio frequency and microwave data collection. The idea of the research is interesting and present enough novelty. The paper title is accurate and concise. The abstract is well written and concise. In the entire manuscript, authors use a standard technical and scientific terminology. The manuscript is very well written and structured. After well written and concise Introduction, authors explained in detailed research problem, methodology and their novel algorithm. The simulation and measurement results were conducted according to the scientifically correct approach. The manuscript topics fit in Sensors journal scope, especially in Sensor technology and applications, remote sensors, as well as, signal processing. I recommended this paper to be accepted after minor revisions.

 Point 1: The paper has some typos and language errors (e.g. line 90 – “Corner et al.” -> “Corner and Lamont”; line 125 – add equation number and renumber equation numbering in entire manuscript; line 129 – renumber reference [46] base on the Instructions for authors; line 147 – “Friedlander et al.” -> “Friedlander and Weiss”; in entire manuscript change “nth” to “n-th”; line 447 – “7 (b) -> “7b”; etc.). 

 Response 1: Fix the typos and language errors.

 2.      Variable name “const” is not adequate for the variable. Please use one letter for a variable name, accordingly to the “Instructions for authors”.

 Response 2: Use “Z” to replace “const”.

 3.      The variable names must have same font style and size in equations, on figures and in the manuscript text. Please describe/introduce all variables used in equations or on figures in the manuscript text.

 Response 3: All variable names have same font style and size in equations. Already described all variables used in equations and figures.

 4.      All known equations must be adequately cited in the entire paper.

 Response 4: All known equations are cited.

 5.      Use MDPI standard font (Palatino Linotype) on figures if you can.

 Response 5: Use Palatino Linotype on figures.

 6.      Please introduce abbreviations if you want to use it abstract or in the manuscript text (e.g. DOA, CRB, FIM etc.).

 Response 6: Already give the full name of abbreviations.

 7.      Please, double check all references and reference style.

 Response 7: Double checked.

 8.      I suggest to slightly expand the Conclusions with main results.

 Response 8: We slightly modify the conclusions.

Sensors EISSN 1424-8220 Published by MDPI AG, Basel, Switzerland RSS E-Mail Table of Contents Alert
Back to Top